# Bone impact after two years of low-dose oral contraceptive use during adolescence

**Lilian Rodrigues Orsolini[1], Tamara Beres Lederer Goldberg[1]\*, Talita Domingues Caldeirão[1], Carla Cristiane da Silva[2], Anapaula da Conceição Bisi Rizzo[1], Talita Poli Biason[1], Altamir Santos Teixeira[3], Helio Rubens Carvalho Nunes[4]**

**1** Postgraduate Program in Tocogynecology, Botucatu School of Medicine, São Paulo State University (UNESP), Botucatu, São Paulo, Brazil, **2** State University of Londrina – UEL, Londrina, Paraná, Brazil, **3** Department of Tropical Diseases and Diagnostic Imaging, Botucatu School of Medicine, São Paulo State University (UNESP), Botucatu, São Paulo, Brazil, **4** Statistical Consultant, Botucatu School of Medicine, São Paulo State University (UNESP), Botucatu, São Paulo, Brazil

\* tamara.goldberg@unesp.br

**Data Availability Statement:** All relevant data are within the paper and its Supporting information files.

## Abstract

### Objective

Data regarding the use and effect of hormonal contraceptives on bone mass acquisition during adolescence are contradictory. The present study was designed to evaluate bone metabolism in two groups of healthy adolescents using combined oral contraceptives (COC).

### Methods

A total of 168 adolescents were recruited from 2014 to 2020 in a non-randomized clinical trial and divided into three groups. The COC1 group used 20 μg Ethinylestradiol (EE)/ 150 μg Desogestrel and the COC2 group used 30 μg EE/3 mg Drospirenone over a period of two years. These groups were compared to a control group of adolescent non-COC users. The adolescents were submitted to bone densitometry by dual-energy X-ray absorptiometry and measurement of bone biomarkers, bone alkaline phosphatase (BAP), and osteocalcin (OC) at baseline and 24 months after inclusion in the study. The three groups studied were compared at the different time points by ANOVA, followed by Bonferroni's multiple comparison test.

### Results

Incorporation of bone mass was greater in non-users at all sites analyzed (4.85 g in lumbar Bone mineral content (BMC)) when compared to adolescents of the COC1 and COC2 groups, with a respective increase of 2.15 g and loss of 0.43g in lumbar BMC (P = 0.001). When comparing subtotal BMC, the control increased 100.83 g, COC 1 increased 21.46 g, and COC 2 presented a reduction of 1.47 g (P = 0.005). The values of bone markers after 24 months are similar for BAP, being 30.51 U/L (± 11.6) for the control group, 34.95 U/L (± 10.8) for COC1, and 30.29 U/L for COC 2 (± 11.5) (P = 0.377). However, when we analyzed OC, we observed for control, COC 1, and COC 2 groups, respectively, 13.59 ng/mL (± 7.3),

**Funding:** This work was supported by FAPESP (Fundação de Amparo à Pesquisa do Estado de São Paulo [Grants 2007/07731-0, 2011/05991-0, and 2015/04040-2]; Pro-Rector for Research at UNESP and UNIMED ASSIS. All the funders had no role in study design, data collection and analysis, decision to publish, or preparation of the manuscript. There was no additional external funding received for this study.

**Competing interests:** The authors have declared that no competing interests exist.

6.44 ng/mL (± 4.6), and 9.48 ng/mL (± 5.9), with P = 0.003. Despite loss to follow-up occurring in the three groups, there were no significant differences between the variables in adolescents at baseline who remained in the study during the 24-month follow-up and those who were excluded or lost to follow-up.

## Conclusion

Bone mass acquisition was compromised in healthy adolescents using combined hormonal contraceptives when compared to controls. This negative impact seems to be more pronounced in the group that used contraceptives containing 30 µg EE.

## Clinical trial registration

http://www.ensaiosclinicos.gov.br, RBR-5h9b3c. "Low-dose combined oral contraceptive use is associated with lower bone mass in adolescents".

## Introduction

Adolescence represents a period of extreme importance in the life of a human. Puberty is a landmark of this phase. In females, puberty is characterized by the acceleration of longitudinal growth, development of secondary sexual characteristics, and the occurrence of menarche. Simultaneously, there is expressive bone mass acquisition [1, 2].

Bone mass accrual begins in the embryonic phase and continues throughout the skeleton during childhood, at differing rates. During puberty a significant increase is observed between Tanner pubertal stages 3 and 4 [1, 3] and a plateau is reached in late puberty, at around 18 years of age [4, 5]. Approximately 92% of the total bone mass is attained before in this period of the second decade of life [6]. The loss of this window of opportunity for increasing bone mass during adolescence has a negative impact on bone health in adulthood and old age [7]. Bone health is influenced by endogenous factors, such as inherited genetic factors and exposure to sex hormones, as well as by exogenous factors, such as physical activity, smoking, and medication use [4, 8, 9].

Adolescence is characterized by the awakening of sexuality, which often implies the onset of sexual activity. In addition, the prescription of contraceptives occurs at increasingly younger ages and they are not only used as a contraceptive method [10].

Literature data regarding the use and effect of hormonal contraceptives on bone mass acquisition during the critical years are contradictory. Hormonal contraceptives seem to negatively interfere with bone mass acquisition when used in adolescence [11, 12]. However, in some studies, contraceptive use apparently did not reduce the rate of bone gain [6], or the differences observed between users and controls were not significant to characterize this effect [13, 14].

Therefore, to determine the effect of combined oral contraceptives (COC) on bone mass in adolescents, the present study was designed to evaluate bone mineral density and the concentrations of bone formation markers in healthy adolescent girls who had used two low-dose COC over a period of two years, and to compare the findings with those observed in healthy adolescent girls who had not used contraceptive methods.

## Materials and methods

This was a non-randomized clinical trial that included healthy adolescent girls aged 12 to 20 incomplete years, recruited from 2014 to 2020. The study is registered under clinical trial

registration number RBR-5h9b3c with the title "Bone mineral density in adolescents using combined oral contraceptives".

The girls were volunteers seen on an outpatient basis and classified as Tanner stages B4 or B5, who already had their first menstruation and who had regular menstrual cycles.

Patients without an indication for use of a contraceptive method because they had no active sexual life were allocated to the control group. For girls with an indication, all contraceptive methods appropriate for this age group were presented by the health professionals and those who chose to use COC were included in the study. The COC methods defined were combinations of 20 μg Ethinylestradiol (EE)/150 μg Desogestrel (COC1 group) and 30 μg EE/3 mg Drospirenone (COC2 group).

All participants were advised and encouraged to use dual protection (male condom concomitantly with the contraceptive method) in order to prevent sexually transmitted infections. In addition, all the girls were healthy, non-smokers and non- alcohol drinkers, who did not use illicit drugs or medications such as anticonvulsants, anticoagulants, antiretroviral agents, antacids containing aluminum, corticosteroids, or calcium or iron supplements that could interfere with bone mass gain. The participants did not practice sports outside the school and the participation in school sports did not exceed 2 h per week. Adolescents with chronic renal, gastrointestinal, or endocrine diseases, such as diabetes mellitus, early or late puberty, or polycystic ovary syndromes and those with a history of oral contraceptive use or pregnancy were not eligible.

Written informed consent was obtained from all individual participants included in the study or their legal representative for authorization and participation in the study. The project was submitted to the Research Ethics Committee of the Botucatu School of Medicine, São Paulo, Brazil (ethical clearance certificate number 52928416.6.0000.5411) and an amendment was approved by Plataforma Brasil under number 2.766.807.

Anthropometric data were collected and secondary sexual characteristics were evaluated by visual inspection of the breasts and pubic hair and classified according to the Tanner criteria [2]. The adolescents participating in the study were healthy and had a height and body mass index (BMI) between the 5th and 95th percentile for each age group according to the Centers for Disease Control and Prevention growth charts [15].

Bone age (BA) for evaluation of the degree of skeletal maturation was obtained from all adolescents using the method of Greulich & Pyle. The data were interpreted by a single trained evaluator who was unaware of the group to which the adolescent belonged (evaluator blinding).

Bone mineral density (BMD) was evaluated in all adolescents (controls and COC users) at the time of inclusion in the study and after 12 and 24 months of follow- up by dual-energy X-ray absorptiometry (DXA), using a Hologic QDR 4500 Discovery A densitometer (Hologic Inc., Bedford, MA). Lumbar spine (L1-L4) and total and subtotal (without head segment) BMD measurements were obtained [16]. All assessments were performed by a trained professional, who was unaware whether or not the adolescent used a COC (evaluator blinding).

Blood samples were collected by venipuncture, in the morning after a 10-hour fast, and centrifuged for 15 min at 1,500 *g* for the separation of serum. The samples were stored at -70oC until the time of analysis of the biomarkers [bone alkaline phosphatase (BAP) and osteocalcin]. Osteocalcin and BAP were measured using the MicroVue Enzyme Immunoassay (EIA) (Quidel®, San Diego, CA, USA). This immunoassay is a competitive ELISA test that quantifies only intact osteocalcin (ng/mL) as an indicator of bone turnover and does not detect fragments of reabsorbed bone tissue. The intra- and inter-assay coefficients of variation obtained as a measure of precision of the assay ranged from 5 to 10%, as recommended by the manufacturer. For BAP, expressed in U/L, the intra-assay coefficient of variation ranged from 4 to 6% and the inter-assay coefficient of variation ranged from 5 to 8%.

All adolescents were evaluated at intervals of 3 months during the proposed follow-up period of two years. On this occasion, the continuity of use of the prescribed COC was analyzed, as well as the permanence of the adolescent in the study according to the strict criteria proposed for their inclusion. In the case of adolescents who did not appear on the scheduled day for densitometry or blood collection, new appointments were offered close to the days defined for the examinations. If the adolescents did not show up on any of the three scheduled days for the examinations at each proposed time point, or if the results obtained at 12 or 24 months were incomplete, the adolescent was included in the statistical analyses up to her participation in the follow-up. The same care was provided to all adolescents who continued in the follow-up for the proposed period, according to the guidelines of the health services they attended.

## Statistical analysis

First, the homogeneity of the groups was verified. There was no significant violation of the theoretical assumptions of normality of the residues (through the Shapiro-Wilk test and histograms) and homoscedasticity (through the Levene test and dispersion between residuals and predictions of the models), corroborating the adopted models.

Comparisons between groups regarding anthropometric, densitometric, and bone marker variables at baseline and after 24 months involving all selected participants were performed using the ANOVA model with fixed effects, followed by the Bonferroni test for multiple comparisons.

The comparisons between participants who remained in the study and participants who left the study in relation to the variables at baseline in each of the groups were performed using the Student's t test.

The comparisons between the groups in relation to the evolution (difference between the moments 24 months and baseline) of the outcomes (Lumbar BMD_0 to 24, Lumbar BMC_0 to 24, Total Body BMD_0 to 24, Total Body BMC_0 to 24, Subtotal BMD_0 to 24, Subtotal BMC_0 to 24, Fat Mass_0 to 24, Osteocalcin_0 to 24, BAP_0 to 24) were performed by fitting multiple linear regression including basal bone age, BMI, and Total body BMD as adjustment variables.

The groups were compared at baseline only among participants who completed 24 months, using the ANOVA model with fixed effects followed by Bonferroni.

Differences or relationships in the regression models were considered statistically significant if $P < 0.05$. Analyses were performed using SPSS 21 software.

## Results

A total of 168 adolescents were included. Twelve of the 31 adolescents of the control group were lost to follow up for different personal reasons. Twenty-three of the 55 adolescents included in the COC1 group and 34 of the 82 included in the COC2 group completed two years of COC use (Fig 1).

At baseline, no significant differences in chronological age, bone age, or lumbar spine or subtotal body densitometric values were observed between the three groups. The mean age was 15.3 years in the control group, 15.8 years in the COC1 group, and 15.8 years in the COC2 group (p = 0.294) (Table 1).

Regarding the anthropometric data, users of EE/Drospirenone (COC2) were in the 62nd percentile for BMI, a value slightly higher than that observed for users of EE/Desogestrel (COC1) (54.1th percentile). Both groups were situated in higher percentiles than the controls (53.3th percentile) but the difference was not statistically significant (p = 0.065). Analysis of

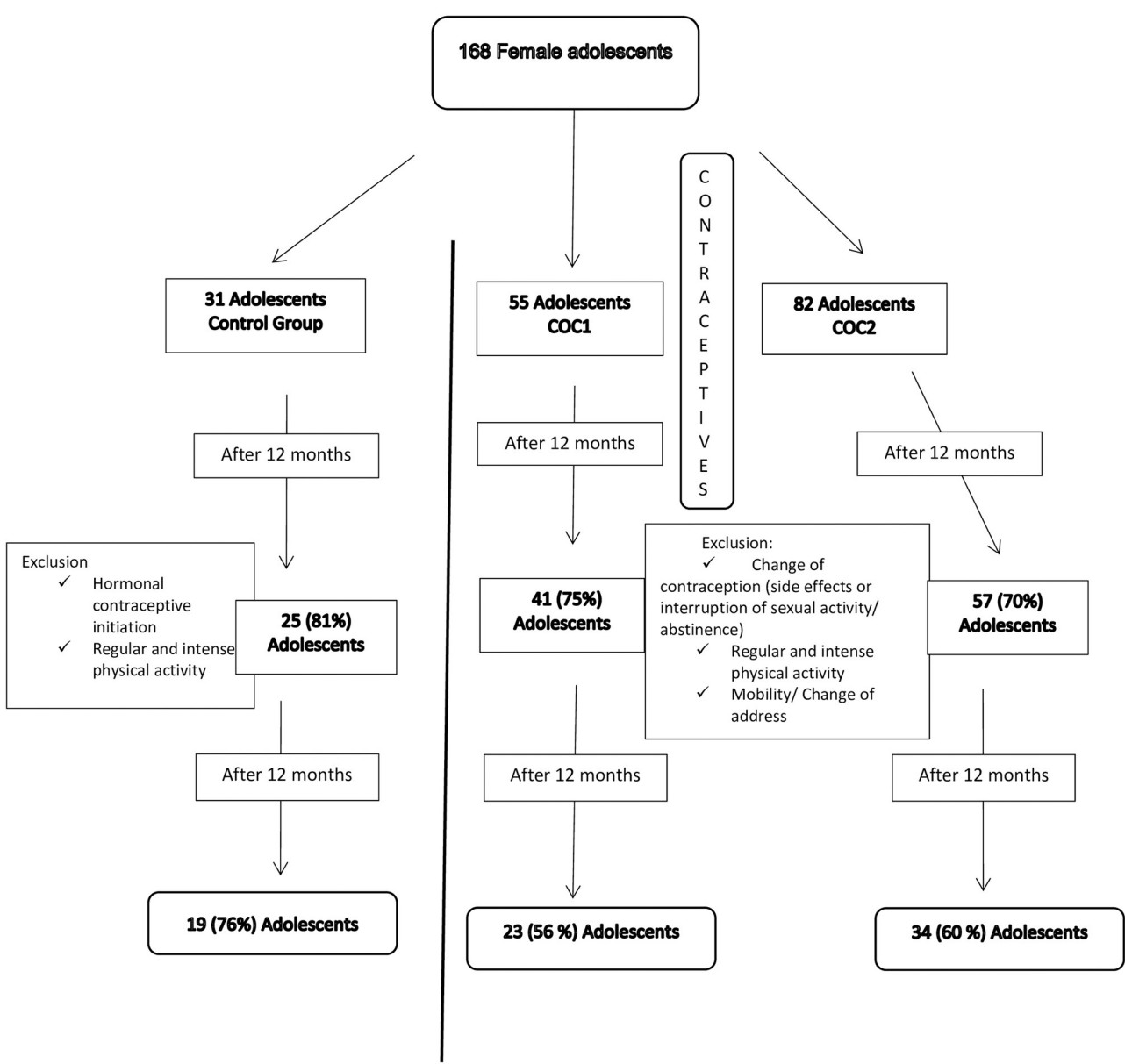

**Fig 1. Flow of participants in the control and contraception groups over 24 months.** COC1: adolescents receiving an oral contraceptive containing 20 μg ethinylestradiol/150 μg desogestrel; COC2: adolescents receiving an oral contraceptive containing 30 μg ethinylestradiol/3 mg drospirenone.

the densitometric measurements showed similar lumbar spine bone mineral content (BMC) and BMD in the three groups (p = 0.119 and p = 0.133, respectively). The three groups were also homogenous in terms of subtotal BMD and BMC (p = 0.063 and p = 0.305, respectively) (Table 1).

With respect to bone markers at baseline, there was no difference in BAP concentrations between the three groups (control: 44.49±22.10 U/L; COC1: 49.21± 25.96U/L, and COC2: 41.58± 16.15 U/L) (p = 0.172). In contrast, osteocalcin concentrations were significantly higher (p<0.05) at baseline in non-users (15.69±7.69 ng/mL) when compared to COC users (COC1: 9.94± 6.03ng/mL and COC2: 10.41±6.26 ng/mL).

**Table 1. Comparison of anthropometric and densitometric variables and bone formation markers at baseline between adolescents receiving low-dose oral contraceptives and the control group.**

| | Controls (n = 31) Mean ± SD | COC1 (n = 55) Mean ± SD | COC2 (n = 82) Mean ± SD | P Value |
|---|---|---|---|---|
| Age (years) | 15.3 ± 1.9 | 15.8 ± 1.8 | 15.8 ± 1.5 | 0.294 |
| Bone age (years) | 15.7 ± 1.8 | 16.2 ± 1.3 | 16.5 ± 1.1 | 0.056 |
| Weight (kg) | 53.9 ± 8.6 | 53.4 ± 7.9 | 55.7 ± 8.0 | 0.228 |
| Height (cm) | 160.9 ± 5.3 | 159.2 ± 5.8 | 159.0 ± 6.4 | 0.309 |
| BMI (kg/m$^2$) | 20.7[a] ± 2.8 | 20.9[a] ± 2.6 | 21.9[b] ± 2.7 | **0.028** |
| Z-score for BMI | 0.1 ± 0.7 | 0.1 ± 0.8 | 0.4 ± 0.7 | 0.064 |
| BMI (percentile) | 53.3 ± 24.6 | 54.1 ± 25.8 | 62.7 ± 23.5 | 0.065 |
| Lumbar BMD (g/cm$^2$) | 0.902 ± 0.090 | 0.955 ± 0.155 | 0.951 ± 0.114 | 0.133 |
| Lumbar BMC (g) | 47.20 ± 8.71 | 50.71 ± 8.26 | 50.80 ± 8.68 | 0.119 |
| Z-score for lumbar | -0.3 ± 0.8 | -0.2 ± 1.0 | -0.2 ± 1.1 | 0.786 |
| Total body BMD (g/cm$^2$) | 1.138[a] ± 0.081 | 0.995[b] ± 0.079 | 1.015[b] ± 0.076 | **0.000** |
| Total body BMC (g) | 1,973.27[a] ± 290.59 | 1,806.94[b] ± 235.21 | 1,835.79[b] ± 253.12 | **0.015** |
| Z-score for total body | 1.5[a] ± 1.0 | -0.7[b] ± 1.2 | -0.5[b] ± 1.0 | **0.000** |
| Subtotal BMD (g/cm$^2$) | 0.905 ± 0.062 | 0.870 ± 0.063 | 0.893 ± 0.073 | 0.063 |
| Subtotal BMC (g) | 1,339.08 ± 227.51 | 1,388.21 ± 188.34 | 1,408.52 ± 214.40 | 0.305 |
| Fat mass (g) | 17,678.2[ab] ± 4,244.2 | 16,178.5[a] ± 5,522.4 | 18,524.6[b] ±4,782.5 | **0.048** |
| Lean mass (g) | 34,188.9 ± 5,174.3 | 36,878.8 ± 4,394.6 | 35,846.3 ± 6,344.3 | 0.127 |
| Total body fat (%) | 32.7[a] ± 4.5 | 28.5[b] ± 5.0 | 32.3[a] ± 5.0 | **0.000** |
| BAP (U/L) | 44.49 ± 22.1 | 49.21 ± 25.96 | 41.58 ± 16.15 | 0.172 |
| Osteocalcin (ng/mL) | 15.69[a] ± 7.69 | 9.94[b] ± 6.0 | 10.41[b] ± 6.26 | **0.000** |

*Note*: Controls: adolescents who did not use oral contraceptives.

COC1: adolescents receiving an oral contraceptive containing 20 µg EE/150 µg desogestrel.

COC2: adolescents receiving an oral contraceptive containing 30 µg EE/3 mg drospirenone.

BMI: Body mass index; BMD: Bone mineral density; BMC: Bone mineral content; BAP: Bone alkaline phosphatase;

ANOVA for comparison of means between the three groups.

Different lowercase letters indicate significant differences between the three groups (p<0.05). Bonferroni test for multiple comparisons between the three groups.

After two years of follow-up, the groups did not differ significantly in terms of bone age or anthropometric data (Table 2).

An average bone mass gain in all segments was observed in the control group, with a gain of 4.85 g in lumbar BMC and of 0.051 g/cm2 in lumbar BMD. Different results were obtained for the COC2 group, with a reduction of 0.012 g/cm2 in lumbar BMD and of 0.43 g in lumbar BMC (p = 0.001). An increase in the lumbar spine densitometric parameters was observed among users of EE/Desogestrel (COC1) but this increase was lower than that detected in adolescents of the control group (2.15 g in lumbar BMC and 0.019 g/cm2 in lumbar BMD; p = 0.258 and p = 0.342, respectively). Analysis of total body BMD and BMC after 24 months showed an increase in the control group but not in the groups of COC users (p<0.05). Subtotal BMC decreased in the COC2 group after 24 months (reduction of 1.47 g), while the control group exhibited an increase of 100.83 g and the COC1 group of 21.56 g over the same follow-up period (p = 0.005) (Fig 2).

With respect to bone markers (BAP and osteocalcin), there was a similar significant decrease in the three groups studied, with no significant difference in BAP (p = 0.686) or osteocalcin (p = 0.909) over the follow-up period of 24 months (Fig 2).

When performing the analysis of multiple linear regression adjusted for adjustment variables BA, BMI, and total body BMD at baseline, shown in S1 Table, the results corroborate those shown in Fig 2.

**Table 2. Comparison of anthropometric and densitometric variables and bone formation markers after 24 months between adolescents receiving low-dose oral contraceptives and the control group.**

| | Controls (n = 19) Mean ± SD | COC1 (n = 23) Mean ± SD | COC2 (n = 34) Mean ± SD | P Value |
|---|---|---|---|---|
| Age (years) | 17.4 ± 2.0 | 18.0 ± 2.0 | 17.6 ± 1.5 | 0.358 |
| Bone age (years) | 17.1 ± 0.9 | 17.5 ± 1.1 | 17.2± 0.9 | 0.610 |
| Weight (kg) | 59.1 ± 9.4 | 56.3 ± 10.5 | 56.9 ± 8.5 | 0.588 |
| Height (cm) | 161.9 ± 4.9 | 160.3 ±6.3 | 159.7 ± 7.5 | 0.481 |
| BMI (kg/m$^2$) | 22.5 ± 3.1 | 21.8 ± 3.5 | 22.1 ± 2.8 | 0.756 |
| Z-score for BMI | 0.4 ± 0.6 | -0.03 ± 0.9 | 0.2 ± 0.6 | 0.154 |
| BMI (percentile) | 64.4 ± 22.2 | 49.7 ± 27.4 | 57.2 ± 21.9 | 0.169 |
| Lumbar BMD (g/cm$^2$) | 0.950 ± 0.070 | 0.932 ± 0.083 | 0.932 ± 0.111 | 0.790 |
| Lumbar BMC (g) | 52.31 ± 8.31 | 50.64 ± 7.89 | 50.24 ± 8.30 | 0.683 |
| Z-score for lumbar | -0.4 ± 0.7 | -0.7 ± 0.9 | -0.7 ± 1.1 | 0.494 |
| Total body BMD (g/cm$^2$) | 1.225[a] ± 0.081 | 0.980[b] ± 0.044 | 1.031[c] ± 0.073 | <**0.001** |
| Total body BMC (g) | 2,276.84[a] ± 245.26 | 1,792.57[b] ± 188.81 | 1,829.13[b] ± 255.50 | **0.000** |
| Z-score for total body | 2.0[a] ± 1.1 | -1.4[b] ± 0.6 | -0.8[b] ± 1.0 | **0.000** |
| Subtotal BMD (g/cm$^2$) | 0.937[a] ± 0.051 | 0.849[b] ± 0.039 | 0.898[c] ± 0.061 | **0.000** |
| Subtotal BMC (g) | 1,467.10 ± 181.33 | 1,372.95 ± 177.97 | 1,382.38 ± 200.73 | 0.230 |
| Fat mass (g) | 22,306.7 ± 5,411.8 | 18,515.1 ± 6,165.9 | 19,291.6 ± 5,533.0 | 0.096 |
| Lean mass (g) | 33,624.9 ± 5,328.4 | 37,249.8 ± 5,781.4 | 36,107.0 ± 4,327.3 | 0.083 |
| Total body fat (%) | 38.2[a] ± 4.8 | 31.5[b] ± 5.0 | 33.1[b] ± 5.1 | <**0.001** |
| BAP (U/L) | 30.51 ± 11.6 | 34.95 ± 10.8 | 30.29 ± 11.5 | 0.377 |
| Osteocalcin (ng/mL) | 13.59[a] ± 7.3 | 6.44[b] ± 4.6 | 9.48[ab] ± 5.9 | **0.003** |

Note: Controls: adolescents who did not use oral contraceptives.

COC1: adolescents receiving an oral contraceptive containing 20 μg EE/150 μg desogestrel.

COC2: adolescents receiving an oral contraceptive containing 30 μg EE/3 mg drospirenone.

BMI: Body mass index; BMD: Bone mineral density; BMC: Bone mineral content; BAP: Bone alkaline phosphatase

ANOVA for comparison of means between the three groups. Different lowercase letters indicate significant differences between the three groups (p<0.05). Bonferroni test for multiple comparisons between the three groups.

It should be highlighted that there were no significant differences in the mean baseline values of each variable analyzed between the adolescents who remained in the study during the 24-month follow-up and those who were excluded or lost to follow- up (Table 3 and S2 Table).

## Discussion

The present study demonstrated significant differences in the evolution of densitometric parameters between healthy adolescent non-COC users and the two groups of COC users over the same two-year follow-up period. In contrast to adolescents of the COC1 and COC2 groups, those of the control group exhibited an increase in BMD and BMC at all sites analyzed (lumbar, subtotal, and total body). In particular, a reduction in lumbar BMC and BMD was observed in the COC2 group. These results indicate that adolescent COC users do not exhibit the same bone mass acquisition during a period of life that is considered extremely important for bone acquisition, compromising the window of opportunity and possibly resulting in irreversible damage in future years [3].

A Canadian prospective multicenter study included young adults and adolescents. Despite its multicenter design, the final sample did not differ from the number of adolescents included in the current study (168 adolescents). The participants were followed up for two years and

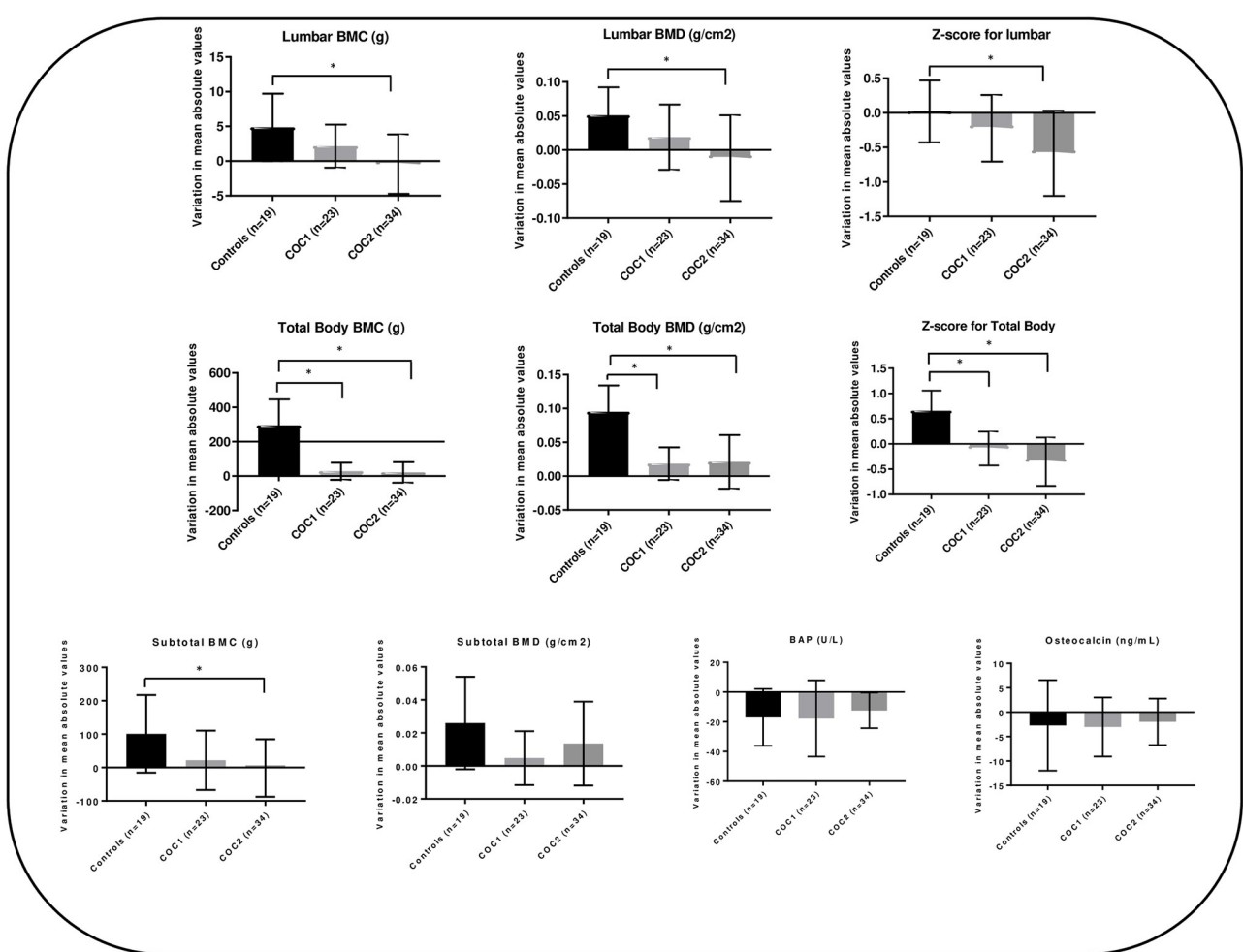

**Fig 2. Comparison of the variation in densitometric variables and bone markers between adolescents followed up for 24 months.** * p<0.05.
ANOVA for comparison of means between the three groups. Bonferroni test for multiple comparisons between the three groups.

COC users showed lower BMD gain than controls, but the difference was not significant.
Losses to follow-up over time and different combinations and doses of the hormonal compo-
nents may explain the nonsignificant and less relevant results compared to those observed in
our study [14].

The results of the present study corroborate data of a recent systematic review with meta-
analysis regarding the lumbar site [17]. A significant increase in total body bone mass was
observed in the control group compared to the lower-than-expected changes in the groups of
COC users (Fig 2). Data referring to the total body segment were not analyzed in that review
because of the heterogeneity of the few published studies [18, 19]. The meta-analysis included
only five studies classified as good quality and the authors highlighted the difficulties faced by
various investigators in following up adolescents using contraceptive methods for a long
period. An important heterogeneity of the studies was that the control group was initially
younger than the users [18] and the fact that young adults and adolescents who had previously
used COC were not excluded from the analysis of the results [19], which almost certainly influ-
enced the odds ratio of the meta-analysis.

**Table 3. Comparison of baseline anthropometric and densitometric variables and bone formation markers between adolescents who remained in the study and those lost to follow-up.**

| | Controls (n = 31) | | | COC1 (n = 55) | | | COC2 (n = 82) | | |
| --- | --- | --- | --- | --- | --- | --- | --- | --- | --- |
| | Remained in the study (n = 19) | Lost to follow-up (n = 12) | | Remained in the study (n = 23) | Lost to follow-up (n = 32) | | Remained in the study (n = 34) | Lost to follow-up (n = 48) | |
| *Variables* | Mean ± SD | Mean ± SD | *P value* | Mean ± SD | Mean ± SD | P value | Mean ± SD | Mean ± SD | *P value* |
| Age (years) | 15.4 ± 2.0 | 15.0 ± 1.8 | 0.591 | 15.9 ± 1.7 | 15.7 ± 1.9 | 0.682 | 15.5 ± 1.4 | 15.9 ± 1.5 | 0.147 |
| Bone age (years) | 15.8 ± 1.9 | 15.8 ± 1.7 | 0.977 | 16.8 ± 0.9 | 16.0 ± 1.3 | 0.190 | 16.3 ± 1.1 | 16.5 ± 1.1 | 0.442 |
| Weight (kg) | 54.9 ± 8.4 | 52.2 ± 9.1 | 0.389 | 52.7 ± 7.9 | 53.8 ± 7,9 | 0.634 | 54.3 ± 8.4 | 56.6 ± 7.7 | 0.197 |
| Height (cm) | 161.2 ± 5.0 | 160.5 ± 6.0 | 0.709 | 159.9 ± 6.5 | 158.7 ± 5.4 | 0.446 | 157.7 ± 6.98 | 159.9 ± 5.8 | 0.112 |
| BMI (kg/m$^2$) | 21.1 ± 2.9 | 20.15 ± 2.4 | 0.387 | 20.6 ± 2.6 | 21.2 ± 2.6 | 0.435 | 21.7 ± 2.8 | 22.2 ± 2.7 | 0.490 |
| Z-score for BMI | 0.3 ± 0.6 | -0.17 ± 0.8 | 0.067 | -0.06 ± 0.7 | 0.2 ± 0.8 | 0.244 | 0.3 ± 0.6 | 0.4 ± 0.8 | 0.587 |
| BMI (percentile) | 60.4 ± 21.6 | 44.8 ± 26.2 | 0.079 | 48.9 ± 24.7 | 56.7 ± 26.4 | 0.301 | 61.5 ± 22.0 | 63.7 ± 24.7 | 0.681 |
| Lumbar BMD (g/cm$^2$) | 0.899 ± 0.090 | 0.907 ± 0.094 | 0.816 | 0.895 ± 0.060 | 0.977 ± 0.173 | 0.100 | 0.944 ± 0.111 | 0.954 ± 0.117 | 0.714 |
| Lumbar BMC (g) | 47.46 ± 9.12 | 46.84 ± 8.46 | 0.847 | 47.44 ± 5.31 | 51.89 ± 8.86 | 0.097 | 50.23 ± 8.46 | 51.16 ± 8.89 | 0.655 |
| Z-score for lumbar | -0.4 ± 0.8 | -0.2 ± 0.8 | 0.627 | -0.6 ± 0.8 | -0.06 ± 1.06 | 0.073 | -0.07 ± 1.0 | -0.2 ± 1.2 | 0.578 |
| Total body BMD (g/cm$^2$) | 1.130 ± 0.073 | 1.151 ± 0.094 | 0.496 | 0.962 ± 0.052 | 1.007 ± 0.085 | 0.095 | 1.013 ± 0.077 | 1.016 ± 0.076 | 0.858 |
| Total body BMC (g) | 1982.04 ± 277.09 | 1959.39 ± 322.97 | 0.837 | 1752.11 ± 218.65 | 1826.87 ± 241.04 | 0.352 | 1790.76 ± 252.68 | 1857.41 ± 252.00 | 0.292 |
| Z-score for total body | 1.4 ± 0.9 | 1,7 ± 1.2 | 0.305 | -1.3 ± 0.6 | -0.5 ± 1.3 | 0.060 | -0.5 ± 1.0 | -0.5 ± 1.0 | 0.818 |
| Subtotal BMD (g/cm$^2$) | 0.911 ± 0.051 | 0.897 ± 0.077 | 0.560 | 0.840 ± 0.045 | 0.881± 0.066 | 0.055 | 0.887 ± 0.062 | 0.895 ± 0.079 | 0.675 |
| Subtotal BMC (g) | 1366.27 ± 214.76 | 1296.0 ± 249.8 | 0.412 | 1338.47 ± 173.13 | 1406.30 ± 192.88 | 0.291 | 1376.6 ± 195.55 | 1423.84 ± 223.14 | 0.379 |
| Fat mass (g) | 18666.3 ± 4330.4 | 16113.8 ± 3753.1 | 0.104 | 18088.8 ± 7358.3 | 15483.8 ± 4632.0 | 0.164 | 17611.7 ± 5357.2 | 18924.0 ± 4510.4 | 0.298 |
| Lean mass (g) | 34590.2 ± 4822.8 | 33586.9 ± 5289.1 | 0.612 | 36692.2 ± 4824.5 | 36948.7 ± 4302.4 | 0.865 | 34345.6 ± 8136.3 | 36531.4 ± 5299.1 | 0.195 |
| Total body fat % | 33.8 ± 4.3 | 31.0 ± 4.4 | 0.098 | 29.9 ± 4.3 | 28.07 ± 5.2 | 0.284 | 31.6 ± 5.2 | 32.7 ± 4.9 | 0.403 |
| BAP (U/L) | 47.07 ± 23.9 | 40.6 ± 19.5 | 0.444 | 51.6 ± 33.7 | 48.46 ± 23.6 | 0.731 | 41.9 ± 15.9 | 41.46 ± 16.4 | 0.916 |
| Osteocalcin (ng/mL) | 16.6 ± 8.1 | 15.0 ± 1.8 | 0.409 | 9.46 ± 5.5 | 10.1 ± 6.3 | 0.715 | 11.08 ± 6.1 | 10.19 ± 6.4 | 0.613 |

*Note*: BMI: Body mass index

BMD: Bone mineral density

BMC: Bone mineral content

BAP: Bone alkaline phosphatase

Student's t-test

Despite the difference in design compared to the present study, Cibula et al., in a prospective cross-over study with a control group and two groups of users (EE 30 and 15 μg), showed that the effect of contraceptives on the increase in bone mass was negative, preventing users from obtaining the expected bone acquisition, as observed in controls [12].

It should be noted that our study design is based on strict inclusion criteria for all adolescents in the sample. This fact can increase the risk of loss to follow-up over a long observation

period, in this case 24 months, which was observed in the three groups studied. The CHOICE study highlights that only 46.7% of adolescents aged 14 to 19 years continue to use hormonal contraceptives for more than 6 to 12 months [20]. Considering factors such as personal decisions, interruption of sexual activity, or side effects of contraceptives that interfere with the discontinuation of the chosen and prescribed method, it is noteworthy that in the present study more than 70% of adolescents continued in the groups after 12 months of follow-up (Fig 1).

Despite the high percentage of losses in the groups after 24 months of follow-up described above, an analysis between the adolescents who remained in the study compared to those who were excluded showed no difference between the parameters studied, ruling out interference or bias in the results.

Assessment of BMD is a static parameter and, therefore, does not reflect dynamic changes in bone tissue occurring at the time of measurement or shortly before. Bone resorption and formation are intimately linked processes in bone remodeling and estrogen is an important regulator of both processes [21, 22]. Both estradiol and EE act on estrogen receptors through the same biological mechanisms and EE has been recognized to exert a more potent effect on target tissues [23]. However, the oral route of EE administration implies the hepatic first-pass effect and a consequent reduction in IGF-1, a hormone also essential in the acquisition of bone mass in adolescence. Furthermore, EE results in an increase in sex hormone-binding globulin (SHBG), decreasing the bioavailability of estradiol [24]. These effects possibly collaborate in the reduction in bone mass deposit in adolescents using COC. The present study showed greater involvement of EE 30mcg in the increase in bone mass in users, supporting the hypothesis that the effect on SHBG is dose dependent.

However, Gargano et al. did not identify differences in the effects of formulations containing EE, 20 or 30 mcg, both with added Drospirenone, on bone metabolism [25]. Progesterone exerts an osteoanabolic effect, stimulating bone formation in women with normal estrogen levels [26]. The present study investigated the effect of two different progestogens, with Drospirenone being a derivative of 17-spironolactone and a potent progestogenic with antiandrogenic and antimineralocorticoid activity and Desogestrel, a third generation progestin derived from 19-nortestosterone, which activates androgen receptors by competitive inhibition, thus blocking endogenous androgenic action. It is promulgated that the effect of progestogen on bone metabolism may be associated with the combination of estrogenic component used, either 17 β- Estradiol or EE, as well as with the oral or transdermal route of administration, since a study conducted by Hadji et al. demonstrated no impact on fracture risk in users of contraceptives containing isolated progestin, demonstrating bone mass preservation [22].

Comparing different progestogens, Nappi et al. observed a greater reduction in bone marker concentrations in the Drospirenone group compared to the Gestodene group, a progestin of the same generation as the Desogestrel used in our study [27]. However, our results did not present the same response.

Callegari et al. evaluated young women aged 16 to 25 years and identified the influence of contraceptives on markers of bone formation (P1NP) and resorption (carboxy-terminal telopeptide, CTX). The results showed concentrations 22% lower than those observed in non-users of hormonal contraceptives [28]. A literature review conducted by Herrmann et al. supports the above findings; however, most of the included articles did not exclusively analyze adolescents [29]. Despite this observation, the authors were unable to provide evidence of the influence of these findings on the fracture risk in adolescent girls. In contrast, Lattakova et al. did not detect the same impact of COC use on markers of bone formation (osteocalcin) and resorption (CTX) over a period of one year [6].

Studies investigating bone markers in adolescence related to the use of contraceptives are still scarce in the scientific literature. Within this context, the results of the present study

intend to contribute to the understanding of bone turnover in these adolescent users. The higher concentrations of bone formation markers at baseline observed in the control group (Table 1) may be explained by the fact that these adolescents were 6 months younger than the COC users (mean age of 15.3 years). Studies involving healthy adolescents revealed higher concentrations of bone formation markers between 12 and 13 years of age and an important reduction in these concentrations after 16 years, followed by a substantial decline until the end of adolescence [1, 3]. The mean concentrations of the bone markers (BAP and osteocalcin) decreased in the three groups. The reduction in the mean concentrations of the markers was lower in the COC2 group receiving 30 µg EE compared to the control and COC1 groups, but no statistically significant differences were found (p = 0.686 and p = 0.909, respectively). Bone turnover markers may simultaneously reflect longitudinal growth and bone mass acquisition, which results in interpretation difficulties when adolescents are evaluated [22].

The current discussion is centered around the question of whether a reduction in the estrogenic component of oral contraceptives would have a negative impact on the bone health of adolescents, including reduced bone metabolism, since some studies point to a not yet defined ideal/physiological concentration of endogenous estrogen (window of action) that would exert an optimal effect on bone remodeling and peak bone mass [30]. However, there is consensus among specialists regarding the prescription of COC that are composed of increasingly lower doses of EE in order to reduce thromboembolic complications [31]. Although challenging, studies involving adolescents who receive different options of available contraceptive methods, including long-acting reversible contraceptives, are necessary to identify the most appropriate oral contraceptive composition and to clarify doubts about the deleterious effects on bone health.

Although the present study presents some limitations, such as the loss to follow- up of some adolescents, an analysis of the power of the test for the comparison of the variation in mean absolute values for the outcomes lumbar spine, total body BMD, total body BMC, subtotal BMC between groups, using ANOVA with fixed effects, presented in Fig 2, showed estimated powers between 0.78 and 0.99, indicating relative adequacy of the analyzed sample size.

Despite the difference in the number of participants between the groups, at the initial moment, the groups demonstrated homogeneity in the majority of the variables analyzed (S2 Table).

Furthermore, multiple linear regression analyses were performed, with adjustments for possible confounders, such as bone age, total body BMD, and BMI at baseline, indicating that statistical differences were maintained (S1 Table), and confirming that the negative impact on bone mass in the COC2 group was more intense than that evidenced in the COC1 group.

## Conclusions

Bone mass acquisition was compromised in healthy adolescents using combined hormonal contraceptives for two years when compared to controls. This negative impact seems to be more pronounced in the group that used contraceptives containing 30 µg EE.

## Supporting information

**S1 Table. Multiple linear regression for the comparison of the evolution of the variables between the groups adjusted by basal bone age, BMI, and total body BMD.**
(DOCX)

**S2 Table. Comparison of anthropometric and densitometric variables and bone formation markers at baseline between adolescents receiving low-dose oral contraceptives and the**

**control group who remained in the study.**
(DOCX)

**S3 Table. Comparison of the variation in mean absolute values of the anthropometric and densitometric variables and bone formation markers between adolescents receiving low-dose oral contraceptives and the control group followed up for 24 months.**
(DOCX)

**S1 File. Trial protocol English version.**
(PDF)

**S2 File. Trial protocol Portuguese version.**
(PDF)

**S1 Checklist. TREND statement checklist.**
(PDF)

**S1 Data.**
(XLSX)

## Acknowledgments

The authors thank Prof. Cilmery Suemi Kurokawa and the technicians Maria Regina Moretto (MSc) and Marcia Tenorio Delneri (MSc) of the Center for Pediatric and Experimental Research, Botucatu Medical School, Universidade Estadual Paulista.

## Author Contributions

**Conceptualization:** Lilian Rodrigues Orsolini, Tamara Beres Lederer Goldberg, Talita Domingues Caldeirão, Carla Cristiane da Silva, Anapaula da Conceição Bisi Rizzo, Talita Poli Biason, Altamir Santos Teixeira, Helio Rubens Carvalho Nunes.

**Data curation:** Lilian Rodrigues Orsolini, Talita Domingues Caldeirão, Carla Cristiane da Silva, Anapaula da Conceição Bisi Rizzo, Altamir Santos Teixeira.

**Formal analysis:** Helio Rubens Carvalho Nunes.

**Funding acquisition:** Lilian Rodrigues Orsolini, Tamara Beres Lederer Goldberg.

**Investigation:** Lilian Rodrigues Orsolini, Tamara Beres Lederer Goldberg, Talita Domingues Caldeirão, Carla Cristiane da Silva, Anapaula da Conceição Bisi Rizzo, Talita Poli Biason, Helio Rubens Carvalho Nunes.

**Methodology:** Tamara Beres Lederer Goldberg, Talita Poli Biason, Altamir Santos Teixeira.

**Project administration:** Tamara Beres Lederer Goldberg, Carla Cristiane da Silva, Talita Poli Biason.

**Resources:** Lilian Rodrigues Orsolini, Talita Domingues Caldeirão, Anapaula da Conceição Bisi Rizzo, Altamir Santos Teixeira.

**Software:** Helio Rubens Carvalho Nunes.

**Supervision:** Tamara Beres Lederer Goldberg.

**Visualization:** Carla Cristiane da Silva.

**Writing – original draft:** Lilian Rodrigues Orsolini, Tamara Beres Lederer Goldberg.

**Writing – review & editing:** Lilian Rodrigues Orsolini, Tamara Beres Lederer Goldberg, Carla Cristiane da Silva, Talita Poli Biason.

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
