## [Decision Letter · Decision Letter 0]

23 Aug 2022

PONE-D-21-39479Bone impact after two years of low-dose oral contraceptive use during adolescencePLOS ONE

Dear Dr. Goldberg,

Thank you for submitting your manuscript to PLOS ONE. After careful consideration, we feel that it has merit but does not fully meet PLOS ONE’s publication criteria as it currently stands. Therefore, we invite you to submit a revised version of the manuscript that addresses the points raised during the review process.

The manuscript has been evaluated by two reviewers, and their comments are available below.

The reviewers have raised a number of major concerns. They request improvements to the reporting of methodological aspects of the study, for example, regarding the concentrations of the contraceptive. The reviewers also note concerns about the statistical analyses presented and request re-analyses be completed.

Could you please carefully revise the manuscript to address all comments raised?

We look forward to receiving your revised manuscript.

Kind regards,

Thomas Tischer

Staff Editor

PLOS ONE

Journal Requirements:

2. Please add the name of the clinical trial registry in which you register the trial.

“This work was supported by FAPESP (Fundação de Amparo à Pesquisa do Estado de São Paulo [Grants 2007/07731-0, 2011/05991-0, and 2015/04040-2]; Pro-Rector for Research at UNESP and UNIMED ASSIS. The funders had no role in study design, data collection and analysis, decision to publish, or preparation of the manuscript.”

Reviewers' comments:

Reviewer's Responses to Questions

**Comments to the Author**

1. Is the manuscript technically sound, and do the data support the conclusions?

Reviewer #1: Partly

Reviewer #2: Partly

2. Has the statistical analysis been performed appropriately and rigorously? 

Reviewer #1: Yes

Reviewer #2: No

3. Have the authors made all data underlying the findings in their manuscript fully available?

Reviewer #1: No

Reviewer #2: No

4. Is the manuscript presented in an intelligible fashion and written in standard English?

Reviewer #1: Yes

Reviewer #2: Yes

5. Review Comments to the Author

Reviewer #1: The present study is a quasi-experimental study evaluating the effect of a low-dose oral contraceptive on DXA in women aged 12 to 20 years. The authors found that oral contraceptive users had a lower BMD when compared with non-users.

The main study concerns are the number of subjects lost to the follow-up (About 44% were lost to the follow-up in the Ethinylestradiol (EE)/150 μg Desogestrel group), the broad age range (with a lower mean age in the control group), and some basal differences among the groups (the group with 30 μg EE/3 mg Drospirenone had a higher total BMD at the baseline).

Specific comments

1) The age range appears wide even though the Tanner stages are B4 to B5. How do the authors explain and justify this age range?

2) How was chosen the combination of oral contraceptive? 20 μg Ethinylestradiol (EE)/150 μg Desogestrel(COC1 group) or 30 μg EE/3 mg Drospirenone (COC2 group)?

3) How Desogestrel compares with Drospirenone? Why have the authors chosen to use different progestogens?

4) Why have the authors used the BMI percentile instead of SD?

5) Please give the definition for the variables: total and subtotal body BMD. Why was the ISCD position not followed?

6) Statistica analysis – why was the DXA not adjusted by the estimated bone age? Or basal BMI and BMD? It could be performed using a generalized linear model.

7) Please move Table 3 and the results presented on lines 284 to 288 to the results section.

Reviewer #2: The manuscript addresses a potentially interesting topic. The collected data are original and rich of information. The employed statistical methods are rather basic and not fully suited for the data at hand. Some detailed comments follow.

1. The droput rate is rather relevant. It is well-known that the complete case analysis could be biased. I am wondering why missing-at-random or a missing-not-at-random assumptions are completely neglected. Please, provide support of the underlying modelling assumptions.

2. Results in table 3 are based on simple t-tests. However, even simple tests are based on quite strong assumptions. As the sample sizes are rather small, those assumptions are hardly tenable. Please, provide evidence that all the assumptions to ensure the reliability of the tests are met.

3. The data are longitudinal in nature. Maybe I miss something, but this fundamental data feature is neglected. Association between repeated measurements must be considered.

4. Please, show data description, as in table 1, for complete cases only.

5. Please, provide evidence that "The assumptions of homoscedasticity and normality were tested using the Levene and Shapiro-Wilk tests, respectively, and the results showed normal distribution".

6. I am also wondering why, with such a rich dataset, a basic ANOVA or simple t-tests are considered only. Confounders may play a role and a regression analysis (for longitudinal data, taking into account missingness) may reveal interesting insights, currently swept under the carpet. Of course, model's assumptions must be checked carefully.

6. PLOS authors have the option to publish the peer review history of their article (what does this mean?). If published, this will include your full peer review and any attached files.

Reviewer #1: No

Reviewer #2: No

---

## [Author Response · Author response to Decision Letter 0]

11 Jan 2023

Dear Dr Tischer and Reviewers

We thank you for your important suggestions that helped improve our manuscript. Some of the suggestions were accepted, while others, although pertinent, have not been included in the text, in which case, an explanation of our point of view has been provided. Attached, please find our replies to the suggestions of the reviewers. We again thank the reviewers for their important comments and the detailed appraisal of our manuscript.

We hope to have clarified all doubts and remain at your disposal for further clarifications. 

Sincerely,

Prof. Tamara Goldberg, Lilian Rodrigues Orsolini, MSc and authors

---

## [Decision Letter · Decision Letter 1]

15 Feb 2023

PONE-D-21-39479R1Bone impact after two years of low-dose oral contraceptive use during adolescencePLOS ONE

Dear Dr. Goldberg,

Thank you for submitting your manuscript to PLOS ONE. After careful consideration, we feel that it has merit but does not fully meet PLOS ONE’s publication criteria as it currently stands. Therefore, we invite you to submit a revised version of the manuscript that addresses the points raised during the review process.

We look forward to receiving your revised manuscript.

Kind regards,

Matt A Price

Academic Editor

PLOS ONE

Additional Editor Comments:

Full disclosure, I am a new editor taking care of your manuscript and have reviewed the comments and content in that context. I don’t yet think your paper is ready, but if you are able to address my additional comments, I feel it will make a good addition to the literature. As you respond to the other reviewer’s comments, please ensure you upload your additional modeling details (see comments) as supplementary materials.

Comments:

Abstract: it would be more valuable to the reader to see some metric that describes the incorporation of bone mass and how it differs across groups, rather than just the p value. The p value tells you nothing about the relationship (except that it is statistically significant)

Abstract: “The three groups exhibited a reduction in the concentrations of bone formation markers, without a significant difference.” This is an oxymoron. If you do not see a significant difference in bone formation markers, than the bone formation markers across these three groups as statistically equivalent. To say “exhibited a reduction” is misleading and incorrect. In the discussion, you may wish to bring up statistical power, and that, had you enrolled more participants, you may have seen a statistical difference in the data. See also the results section and part of the discussion (page 14)– it appears as though you have presented these differences as significant when they are not.

Abstract: no mention of the loss to follow up. This is an important point, I would add a sentence to clarify the high rate of attrition, and that this varied by group. In your reply to the comments, you note that those who dropped out did not differ from those who remained on study (am I interpreting this correctly? I’m referring to your comment “despite the difference in the number of participants between the groups, at the initial moment, the groups demonstrated homogeneity in the majority of the variables analyzed”). Is this covered adequately in your results section?

Abstract, conclusions: As a clinician, I would want to know how big of an effect might I see with my patients who wanted to start birth control. As a public health / policy maker, I would want to weigh this against the value of effective birth control. Given your differential loss to follow up, I think it is safer to claim that there may be differences across the groups, but that your research does not definitively show this.

Materials and Methods: You note how the control group was assigned at the bottom of page 4; how were the two other groups assigned?

Materials and Methods: How do you define lost to follow up? When they stopped taking oral contraception? When they stopped attending study visits?

Methods, Statistical Analysis: Please define your outcome. How do you analyze participants who stay on study, but who stop (or switch) contraceptives?

Methods, Statistical Analysis: Kindly add a little more detail on how you assess any bias due to loss to follow up (e.g., did you compare those retained and analyzed with those enrolled to see if they differed by any variables that might confound or bias your results?). There’s nothing on this here, and I fear it may seriously compromise your modeling.

Last sentence of results “This fact reinforces the results obtained despite a high percentage of losses in the groups after 24 months of follow-up, ruling out interference or bias in the results”. This is discussion, I recommend noting this there, as a strength of your study.

Discussion: you observed a statistically significant difference. Do you also feel this is a clinically significant difference? I feel this is a very important element of your paper that gets limited discussion. The final sentence of the first paragraph in the discussion suggests that may be the case, but I’d like to see more clarity on this from the authors.

Reviewers' comments:

Reviewer's Responses to Questions

**Comments to the Author**

1. If the authors have adequately addressed your comments raised in a previous round of review and you feel that this manuscript is now acceptable for publication, you may indicate that here to bypass the “Comments to the Author” section, enter your conflict of interest statement in the “Confidential to Editor” section, and submit your "Accept" recommendation.

Reviewer #1: All comments have been addressed

Reviewer #2: (No Response)

2. Is the manuscript technically sound, and do the data support the conclusions?

Reviewer #1: Yes

Reviewer #2: Partly

3. Has the statistical analysis been performed appropriately and rigorously? 

Reviewer #1: Yes

Reviewer #2: No

4. Have the authors made all data underlying the findings in their manuscript fully available?

Reviewer #1: Yes

Reviewer #2: No

5. Is the manuscript presented in an intelligible fashion and written in standard English?

Reviewer #1: Yes

Reviewer #2: Yes

6. Review Comments to the Author

Reviewer #1: I thank the authors for the through responses. All my comments have been addressed. I believe that the manuscript is now much improved.

Reviewer #2: Though the authros provided some answers to my questions, there is a general lack of statistical knowledge and this strongly limits the work.

Two aspects still deserve more attention: the missing data mechansim and the longitudinal structure of the data.

1. Missingness cannot be "tested" by comparing the covariates between the retained and the dropout groups. The missing data mechanism refers to the main outcomes, and there is a wide literature on the effects of missing-not-at-random mechanisms. As it stands, the modelling approach may lead to biased estimates.

2. " the longitudinal character has been incorporated in the analysis of the S1 Table, which compares the three groups in relation to the evolution of the variables studied". It is rather unclear how the longitudinal structure is accounted for. Did you consider a random effects model, a fixed effect or...? How did you estimate the parameters? Did you test the underlying modelling assumptions? Is there any time-dependence which needs to be considered?

3. "The assumptions of homoscedasticity and normality were tested using the Levene and Shapiro-Wilk tests, respectively, and the results showed normal distribution"." Please, provide graphical evidence of this result.

7. PLOS authors have the option to publish the peer review history of their article (what does this mean?). If published, this will include your full peer review and any attached files.

Reviewer #1: No

Reviewer #2: No

---

## [Author Response · Author response to Decision Letter 1]

28 Mar 2023

Botucatu, March 25th, 2023

Dear Dr Price and Reviewers

We thank you for your important suggestions that helped improve our manuscript. Some suggestions were accepted, and have been included in the text of the manuscript. A point-by-point response to the comments, including a detailed description of the changes made, is presented below. We again thank the reviewers for their important comments and the detailed appraisal of our manuscript. We hope to have clarified all doubts and remain at your disposal for further clarifications. 

Sincerely,

Prof. Tamara Goldberg, Lilian Rodrigues Orsolini, MSc and authors

---

## [Decision Letter · Decision Letter 2]

4 May 2023

Bone impact after two years of low-dose oral contraceptive use during adolescence

PONE-D-21-39479R2

Dear Dr. Goldberg,

We’re pleased to inform you that your manuscript has been judged scientifically suitable for publication and will be formally accepted for publication once it meets all outstanding technical requirements.

Kind regards,

Matt A Price

Academic Editor

PLOS ONE

Additional Editor Comments (optional):

Reviewers' comments:

Reviewer's Responses to Questions

**Comments to the Author**

1. If the authors have adequately addressed your comments raised in a previous round of review and you feel that this manuscript is now acceptable for publication, you may indicate that here to bypass the “Comments to the Author” section, enter your conflict of interest statement in the “Confidential to Editor” section, and submit your "Accept" recommendation.

Reviewer #2: All comments have been addressed

2. Is the manuscript technically sound, and do the data support the conclusions?

Reviewer #2: (No Response)

3. Has the statistical analysis been performed appropriately and rigorously? 

Reviewer #2: (No Response)

4. Have the authors made all data underlying the findings in their manuscript fully available?

Reviewer #2: (No Response)

5. Is the manuscript presented in an intelligible fashion and written in standard English?

Reviewer #2: (No Response)

6. Review Comments to the Author

Reviewer #2: (No Response)

7. PLOS authors have the option to publish the peer review history of their article (what does this mean?). If published, this will include your full peer review and any attached files.

Reviewer #2: No

---

## [Editor Report · Acceptance letter]

31 May 2023

PONE-D-21-39479R2 

Bone impact after two years of low-dose oral contraceptive use during adolescence 

Dear Dr. Goldberg:

I'm pleased to inform you that your manuscript has been deemed suitable for publication in PLOS ONE. Congratulations! Your manuscript is now with our production department. 

Kind regards, 

on behalf of

Dr. Matt A Price 

Academic Editor

PLOS ONE